# Prevalence of Hepatitis B and Hepatitis C Infections among Incarcerated Individuals in Iran: A Cross-Sectional National Bio-behavioral Study in 2019

**DOI:** 10.3390/pathogens10111522

**Published:** 2021-11-21

**Authors:** Ghobad Moradi, Seyed Moayed Alavian, Fatemeh Gholami, Rashid Ramezani, Leila Ahangarzadeh, Yousef Moradi, Heidar Sharafi

**Affiliations:** 1Social Determinants of Health Research Center, Research Institute for Health Development, Kurdistan University of Medical Sciences, Sanandaj 6617713446, Iran; moradi_gh@muk.ac.ir; 2Department of Epidemiology and Biostatistics, Faculty of Medicine, Kurdistan University of Medical Sciences, Sanandaj 6617713446, Iran; 3Middle East Liver Diseases Center, Tehran 1598976513, Iran; alavian@thc.ir; 4Department of Epidemiology, School of Public Health, Iran University of Medical Sciences, Tehran 1449614535, Iran; gholamifatemeh734@gmail.com; 5Ministry of Health and Medical Education of Iran, Tehran 141994347, Iran; rashidramezani@gmail.com; 6Faculty of Medical Education, Shahid Beheshti University of Medical Sciences, Tehran 1985717443, Iran; liliahangar@gmail.com

**Keywords:** hepatitis B, hepatitis C, epidemiology, prison, Iran

## Abstract

**Introduction:** To realize the global goals of eliminating hepatitis B virus (HBV) and hepatitis C virus (HCV) by 2030, it is necessary to monitor the status of disease among target populations and undertake the required interventions. This study is the third round of surveys to determine the prevalence of hepatitis B and C infections among incarcerated individuals in different provinces of Iran. **Methods:** This study was conducted in five provinces of Iran (including Kurdistan, Ardabil, West Azerbaijan, Markazi, and Semnan) in 2019. The subjects of the study were selected from incarcerated people in prisons of all provinces that had not been studied in the previous two rounds of the surveys (in 2015 and 2016) in Iran. In this study, 15 prisons were selected and 2475 incarcerated individuals were enrolled into the study based on the multistage sampling method; the selected subjects were surveyed and their dried blood spot (DBS) samples were collected to test HBsAg and HCV-Ab. In cases with a reactive result for HCV-Ab, an HCV-RNA test was also performed on their serum samples. The relationships between independent variables and outcomes were evaluated via logistic regression. **Results:** Of all participants (2475 subjects) enrolled in the study, 54.18% were selected from northern provinces and 45.82% from the central provinces. The prevalence of HCV-Ab and HBsAg among incarcerated individuals was 5.66% (95% CI: 4.81% to 6.64%) and 2.42% (95% CI: 1.89% to 3.11%), respectively. Among HCV-seropositive individuals, 73.68% (95% CI: 64.70% to 81.01%) had current HCV infection (detectable HCV-RNA). The results showed that histories of imprisonment, drug use, unprotected sexual contact, drug injection, tattooing, and younger age in the first-time drug use in incarcerated individuals significantly increased the risk of HCV transmission. Among these behaviors, drug injection was more likely than other behaviors to result in contracting HCV in incarcerated individuals (OR: 22.91; 95% CI: 14.92–35.18; *p* < 0.001). **Conclusion:** To achieve international and national strategies targeted to eliminate HCV and HBV by 2030, it is necessary to pay special attention to prisons in Iran. It is recommended to continue HBV vaccination of eligible people in prisons. Developing screening and treatment protocols for individuals with HCV infection in prisons can help the country to achieve HCV elimination goals.

## 1. Introduction

Viral hepatitis imposes a significant burden on communities and the health system. As estimated, 71 and 257 million people worldwide are currently infected with hepatitis C virus (HCV) and hepatitis B virus (HBV), respectively, which cause approximately 1.4 million deaths per year (687,000 deaths from HBV and 704,000 deaths from HCV) [1,2]. Given the importance of viral hepatitis, in the 69th World Health Assembly in 2016 which was focused on the global strategies and goals for controlling viral hepatitis, the elimination of HBV and HCV by 2030 was set as a global agenda. To achieve this goal, strategies and measures were set, including a 90% reduction in the incidence of chronic infections (i.e., from 6–10 million cases in 2016 to 0.9 million cases in 2030), and a 65% reduction in annual mortality from chronic hepatitis (i.e., from 1.4 million cases in 2016 to less than 0.5 million cases in 2030). Achieving the above goals will ultimately prevent 7 million deaths associated with these diseases [1].

Prison is one of the places playing an important role in the transmission of infectious diseases. The prevalence of HBV and HCV infections among the population of incarcerated individuals is higher than the general population. Studies showed that the majority of incarcerated individuals have lower socioeconomic status, as compared with the general population, which leads to increased prevalence of high-risk behaviors such as unprotected sexual activities, tattooing, drug use, and the use of shared needles. All of the mentioned factors can increase the risk of HBV and HCV transmission in this population [3,4]. According to previous studies, the highest and the lowest prevalence of HBV infection were observed among prisoners in western and central Africa (23.5%) and North America (1.4%), respectively, and in other areas, it varies from 2.3% to 10.4%. In addition, the highest and the lowest prevalence of HCV infection were observed among prisoners in Asia and Pacific (20.6%) and Eastern and Southern Africa (1.8%), respectively, and in other areas, it varies from 4.7% to 20.2% [3].

Moreover, a small number of scattered studies in Iran have been conducted on high-risk groups of the community. In a study in Iran, the prevalence of HBV infection among high-risk groups, including prisoners, was 3.06% [5]. On the other hand, a systematic review study showed that the prevalence of HCV infection among prisoners was 28% [6]. Various interventions have been adopted to reduce the incidence of HBV infection, including vaccination and other successful interventions in this field, however, the prevalence of HCV infection among high-risk groups of the society is still high and requires special measures and interventions. Conducting national and comprehensive studies can help to formulate these measures and implement them for high-risk groups, especially prisoners. Given that the World Health Organization’s most important strategy for eliminating HBV and HCV by 2030 is to develop a strong surveillance system for HBV and HCV, especially among high-risk populations such as the prisoners, it is very important to conduct national studies to strengthen the care system targeted toward controlling these infections [7,8,9]. In 2015, the first HBV and HCV biobehavioral survey was implemented on approximately 6000 prisoners in 9 provinces and 25 prisons in Iran, and the results were published [10]. The results of this study showed that the prevalence of HCV exposure was 9.48% and the prevalence of HBV infection was 2.48% in the general prison population [10]. In addition, the second program of the surveillance system was implemented in 2016 on approximately 6400 prisoners in 10 provinces and 29 prisons of the country, and the results were published [5]. In this study, results showed that the prevalence of HCV-Ab and HBsAg was 8.21% and 3.06%, respectively [5].

The present study is the third study related to the HBV and HCV surveillance system program in Iran, which was conducted to determine the prevalence of HBV and HCV infections among incarcerated people in different provinces of Iran. As the advantage of this round over the previous round of studies, HCV-RNA test was used to diagnose current HCV infection in people who are reactive for HCV-Ab.

## 2. Methods

### 2.1. Study Design and Setting, and Sampling Method

This analytical cross-sectional study was conducted in five selected provinces of Iran (Kurdistan, Ardabil, West Azerbaijan, Markazi, and Semnan) in 2019. The subjects were selected from all incarcerated individuals of provinces that had not been evaluated in the previous two rounds (in 2015 and 2016) [5,10]. In this study, 13 prisons and 3 vocational training and occupational therapy camps were selected, from which a total of 2475 incarcerated individuals were enrolled into the study using the multistage sampling method, the details of which were described elsewhere [5].

### 2.2. Measurement Tools and Data Collection 

The data collected in the hepatitis B and C surveillance program consisted of two-part. In the first part, a designed questionnaire was used to collect the required data related to the demographic characteristics of incarcerated individuals, including the patterns of high-risk behaviors. The validity and reliability of the questionnaire for measuring high-risk behaviors were calculated in previous studies [11,12]. The second step was designed to collect data from laboratory testing. Trained people were recruited to collect data using a questionnaire.

### 2.3. Blood Sampling and Laboratory Tests

The participants were invited for a collection of dried blood spot (DBS) samples. The collected DBS samples were tested for HBsAg and HCV-Ab using enzyme-linked immunosorbent assay (ELISA) and Dia Pro kits (Diagnostic Bioprobes Srl, Milan, Italy). The validity of the tests was ensured using calibrators and positive and negative controls supplied by the manufacturer according to the protocol provided in the manual. Positive and negative samples were determined based on optical density (OD) and cut-off points. In addition, available cases with HCV-Ab were called for anti-coagulated blood sampling to be tested for HCV-RNA using AmpliSens^®^ HCV-FRT PCR kit (Central Research Institute for Epidemiology, Moscow, Russia) with a sensitivity of 50 IU/mL. All the tests were performed in one laboratory.

### 2.4. Statistical Analysis

After collecting the data, first, the normality of the data was checked using the Shapiro–Wilk test. Then, the data were evaluated using descriptive and analytical analyses. The relationship between important factors and the prevalence of HBV and HCV markers was assessed via calculating the odds ratio (95% confidence interval) using univariate logistic regression. In addition, the demographic variables were entered into the adjusted logistic model and the adjusted odds ratio (AOR) was calculated by controlling the important confounding variables. All the analyses were performed in STATA software version 14.

## 3. Results

### 3.1. Characteristics of the Study Participants

A total of 2475 people were evaluated, and their data were analyzed in this study. The mean and standard deviation age of the participants in this study was 37.87 ± 9.67. Of all participants enrolled in the study, 54.18% were selected from northern provinces and 45.82% from the central provinces. In addition, 2418 (97.70%) incarcerated individuals were male, 53.33% were married, and 61.09% had a history of imprisonment. Of all, 57.73% of the participants had a lifetime history of drug use. Moreover, 45.41% of the incarcerated individuals with a lifetime history of drug use practiced drug use for the first time under the age of 20, and 11.65% had a history of drug injection (Table 1).

### 3.2. Prevalence of Hepatitis C Virus Antibodies among Incarcerated Individuals

The prevalence of HCV-Ab in all studied provinces (Kurdistan, Ardabil, West Azerbaijan, Markazi, and Semnan) was 5.66% (95% CI: 4.81% to 6.64%) (Table 2). The prevalence of HCV-Ab among men was 5.75% (95% CI: 4.89% to 6.75%), which was higher than that among women with a prevalence of 1.75% (95% CI: 0.31% to 9.29%), but the difference was not statistically significant (Table 2). The prevalence of HCV-Ab based on other demographic variables was reported in Table 2.

The prevalence of HCV-Ab among incarcerated individuals with a lifetime history of drug use was 8.96% (95% CI: 7.58% to 10.55%), which was higher than that among those without a history of drug use. In this study, 131 participants had a history of opioid-agonist treatment (OAT), and the prevalence of HCV-Ab among these individuals was higher than that among those without a history of OAT (55.20% [95% CI: 44.80% to 62.91%] vs. 34.15% [95% CI: 21.56% to 49.45%]) (Table 2). The prevalence of HCV-Ab among 870 participants with a history of tattooing was 10.57% (95% CI: 8.70% to 12.79%), while the prevalence in those without a history of tattooing was 2.99% (95% CI: 2.26% to 3.94%) (Table 2).

### 3.3. Prevalence of Hepatitis B Surface Antigen among Incarcerated Individuals

The prevalence of HBsAg among incarcerated individuals was 2.42% (95% CI: 1.89% to 3.11%) (Table 2). The prevalence of HBsAg in the central provinces (Semnan and Markazi) was higher than that in the northern provinces (West Azerbaijan, Kurdistan, and Ardabil) (2.56% [95% CI: 1.79% to 3.65%] vs. 2.31% [95% CI: 1.63% to 3.26%]), but the difference was not statistically significant (Table 2). Moreover, the results of this study showed that incarcerated individuals with a lifetime history of drug use had a higher prevalence of HBsAg than those without a history of drug use, but the difference was not statistically significant (2.52% [95% CI: 1.83% to 3.47%] vs. 2.29% [95% CI: 1.55% to 3.39%]). Among participants with a history of drug injection, the prevalence of HBsAg was 3.01% (95% CI: 1.29% to 6.86%), which was higher than that among those without a history of drug injection (2.46%), but the difference was not statistically significant (Table 2). Among illiterate participants, the prevalence of HBsAg was significantly higher than others with various educational levels (4.56% vs. 2.26% and 1.39%) (Table 2).

### 3.4. Modifiers of Hepatitis B Surface Antigen and Hepatitis C Virus Antibodies Prevalence among Incarcerated Individuals

In this study, logistic regression analysis was used to investigate the relationship between demographic variables and high-risk behaviors with the prevalence of HBsAg and HCV-Ab among incarcerated individuals. The results in Table 3 indicate that, through controlling confounding variables, being literate (OR = 0.35) and having a higher educational level (OR = 0.22) were inversely associated with testing reactive for HBsAg (Table 3).

Regarding HCV-Ab prevalence, multivariate logistic regression showed that age ≥ 30 (OR = 6.31), imprisonment longer than a year (OR = 3.87), history of the previous imprisonment (OR = 4.63), history of drug use (OR = 6.45), history of drug injection (OR = 22.91), history of OAT (OR = 2.40), history of tattooing (OR = 3.80), and history of unprotected sexual contact (OR = 1.72) significantly increased the risk of having reactive HCV-Ab test, while having university education (OR = 0.14), being married (OR = 0.30) and age ≥ 20 at the first-time drug use (OR = 0.35) significantly decreased the risk of having reactive HCV-Ab test among incarcerated individuals (Table 3).

### 3.5. Prevalence of Hepatitis C Virus RNA among Incarcerated Individuals

In this study, incarcerated individuals who tested positive for HCV-Ab (N = 140) were invited to undergo HCV-RNA testing. Out of the total number of invited participants, only 114 cases were available for HCV-RNA testing. The results of HCV-RNA testing in these 114 incarcerated individuals showed that 84 (73.68%; 95% CI: 64.70–81.01%) had positive results for HCV-RNA and 30 (26.31%; 95% CI: 18.98–35.24%) had undetectable results for HCV-RNA.

Comparing those with HCV-RNA to those without HCV biomarkers (without HCV-Ab), age, marital status, duration of current imprisonment, history of drug use, age of first-time drug use, history of drug injection, and history of tattooing were independently associated with being positive for HCV-RNA (Table 4). In an additional analysis, the impact of demographic and behavioral parameters on the chronicity of HCV was assessed. In this analysis, age ≥ 30 and ≥1 year current imprisonment were associated with chronicity of HCV infection among incarcerated individuals (Table 5).

## 4. Discussion

This study aimed to determine the prevalence of HBV and HCV infections among Iranian incarcerated individuals. The results showed that the prevalence of HCV-Ab and HBsAg among Iranian incarcerated individuals was 5.66% and 2.42%, respectively. In addition, around 74% of those with HCV-Ab were tested positive for HCV-RNA, indicating the presence of current HCV infection in Iranian incarcerated individuals. According to a systematic review and meta-analysis, the prevalence of HCV-Ab among the general population of Iran was estimated at 0.3%, while in the medium-risk population of Iran, such as incarcerated individuals except for people who inject drugs (PWID), it was 6.2% and in high-risk populations such as PWID, it was estimated to be 32.1% [13]. In this study, the prevalence of HCV-Ab among incarcerated individuals was almost 19 times higher than the prevalence of HCV-Ab among the general population of Iran. Accordingly, the prevalence of HCV-Ab was high in Iranian incarcerated individuals, which is in line with the prevalence reported in a systematic review and meta-analysis [13] conducted on a medium-risk population including incarcerated individuals. Incarcerated individuals with high-risk behaviors suffer from a higher prevalence of HCV infection. Various studies around the world indicated the high prevalence of hepatitis C among incarcerated individuals, especially those with drug injection, which increases the probability of transmission of blood-borne infections such as HCV in prison [14]. In addition, logistic regression analysis showed that history of drug use, history of drug injection, history of tattooing, young age at the first experience of drug use, history of imprisonment, and unprotected sex were the main parameters increasing the risk of having the HCV infection among incarcerated individuals. Hopefully, harm reduction including the needle and syringe program and opioid agonist therapy, was implemented in Iranian prisons since 2005 which was demonstrated to be effective in the containment of HCV epidemics in prisons [15]. In a study conducted in Brazil, the results showed that having a history of drug injection was independently associated with HCV infection [16]. These people are more prone to blood- and non-blood-borne infections such as HCV which is due to their experience of high-risk behaviors in the community and prison. Determining the prevalence of HCV infection among these and other high-risk groups is very important for planning health programs and developing care systems. In addition, the prevalence of HCV and HBV infections among this group leads to improvements in the awareness of health policymakers and health professionals about the status of HCV infection and other infections in prisons, which in turn should lead to health planning and development of services based on the prevalence rates in prisons, because it can prevent uncontrollable pandemics with unfortunate outcomes in prisons and other prison-related places. In the case of the increase in the risk of HCV, the probability of outbreaks of the diseases will increase in prisons, which certainly will not remain in prisons and people can effectively communicate with healthy people in the community after the end of their incarceration. It can lead to the transmission of this disease to the general population of the community [17,18,19]. A study of incarcerated individuals in the Middle East and North Africa region showed pieces of evidence of drug injections and the use of shared non-sterile injection tools in prisons. This study also showed that injecting drug use, non-sterile injection tools, and tattooing in prisons are recognized as independent risk factors for HCV infection, and high levels of risky sexual behaviors, tattooing, and the use of non-sterile blades are among common actions practiced by prisoners [20]. Other studies also suggested that history of drug use and history of imprisonment are known as predisposing factors for HCV [21,22]. Moreover, in a study conducted in Egypt to determine the prevalence of HCV-RNA, tattooing and history of imprisonment were identified as risk factors for developing HCV infection [23]. Tattooing is also an important risk factor for the transmission of HCV in incarcerated individuals. In a prospective two-year study of 320 HCV-Ab-positive cases and 307 HCV-Ab-negative cases in Texas in the Mexico border area, the results showed that tattooing was one of the potential and definitive risk factors for HCV transmission in prisons. In addition to tattooing in the study, drug injection and blood transfusion were recognized as significant independent factors involved in the transmission of hepatitis C infection; it is in line with the results of the present study [24,25,26]. A study of 630 incarcerated individuals in Mexico prison centers showed that drug injection and tattooing were the major risk factors for HCV transmission [27]. The results of a study of 973 incarcerated individuals in eight major Italian prisons indicated that HCV infection was strongly associated with intravenous injections and tattooing [28].

Using HCV-Ab for screening, the prevalence of hepatitis C was higher than the prevalence of the same infection detected via RT-PCR [29]. Since HCV-RNA testing can distinguish between recovered past infection, active infections, and detect HCV-RNA before antibody production, it is considered a more accurate way to determine the prevalence of current HCV infection in high-risk groups, especially incarcerated individuals. The results of the present study showed that not all people who tested positive for HCV-Ab have a positive HCV-RNA test, which could be because not all of them had a current infection. To make plans to implement effective and appropriate programs to achieve WHO targets regarding the elimination of HCV and HBV by 2030, it is necessary to conduct further studies in high-risk populations, including incarcerated individuals. In this study, HCV-RNA was used to diagnose current HCV infection in incarcerated individuals. Anti-HCV tests are the most common screening methods used to diagnose HCV. Despite the advantages of these tests in the detection of antibodies, such as ease of use, low cost, and high sensitivity, recent generations also have some limitations. For instance, it cannot distinguish between current infection and resolved past infection [30]. Based on the above-mentioned facts, HCV-RNA testing together with HCV antibody testing is recommended for the diagnosis of HCV in high-risk groups, especially prisoners [31].

The results of HBV testing using HBsAg detection showed that its prevalence in incarcerated individuals was 2.42%. The World Health Organization estimates that approximately 33% of the world’s population is infected with the HBV, and about 378 million people, i.e., 5% of the world’s population, are chronically infected. Considering the global HBV pattern, Iran is in the category of countries with moderate endemicity. Statistical data have shown that approximately 13.6% of the Iranian population has been exposed to the HBV and about 1.8% of them are chronic carriers [32]. The highest prevalence of HBsAg (60.8%) and HBcAb (85.1%) in Iran was observed in Afghan refugees in a camp in the south of Iran [33]. On the other hand, a meta-analysis has shown that the prevalence of HBsAg among non-injecting people who use drugs and PWID in Iran is 2.9% and 4.8%, respectively [34]. The present study showed that the prevalence of HBsAg in incarcerated individuals with a history of drug injection was 3.01%. In a 12-month study in Victoria, Australia on 3429 male and 198 female incarcerated individuals, 46% were PWID and 2.5% were HBsAg positive [35]. In the mentioned study, a history of unprotected sex was significantly associated with an increased chance of HBV in participants. Currently, unprotected sex is one of the most important routes of transmitting this virus in the world, while drug injection and the use of shared needles is probably the most common route in prisons in Iran [36,37]. In addition, incarcerated individuals with a history of unprotected sex should also be identified to be enrolled in workshops and training programs on how to have protected sex [35,38,39,40].

The crimes and behaviors that lead people to prisons are essentially the same high-risk behaviors that expose a person to infectious diseases. These behaviors include sexual promiscuity, injecting drug use, violence, etc. On the other hand, prisons are not isolated from society. Prisoners are temporarily realized for leave after a short period of imprisonment or are released from prison when they can connect with the community outside the prison. An incarcerated individual is not only exposed to HCV or HBV infection but can also act as a carrier and transmit infectious agents into the community [41].

## 5. Conclusions

According to the results of the present study, it is necessary to continue and promote health interventions and policies to control and prevent HBV and HCV in prisons. To achieve international and national strategies to eliminate HCV and HBV by 2030, it is essential to pay special attention to prisons in Iran. Hence, it is strongly recommended to update diagnostic protocols and implement “HCV test and treat” in prisons [29,42]. Access to people with HCV infection in prisons can be a good opportunity to increase the treatment coverage because of their availability. Treatment of people with HCV in prisons can help to fulfill the goals of controlling and eliminating hepatitis C in Iran. HBV vaccination is still recommended for eligible people in prisons. In addition, the development of screening and treatment protocols for HCV in prisons can help the county to accomplish the hepatitis C control goals.

## Figures and Tables

**Table 1 pathogens-10-01522-t001:** Characteristics and risky behaviors in the incarcerated participants.

Variables	Number (%)	Variables	Number (%)
**Country region**		**Lifetime history of drug use**	
North	1341 (54.18%)	Yes	1429 (57.73%)
Centre	1134 (45.82%)	No	1046 (42.26%)
Total	2475	**Age of drug use onset ^a^**	
**Sex**		<20	649 (45.41%)
Male	2418 (97.70%)	≥20	780 (54.58%)
Female	57 (2.30%)	**Lifetime history of drug injection ^a^**	
**Age group**		Yes	166 (11.65%)
<30	491 (19.84%)	No	1258 (88.35%)
≥30	1984 (80.16%)	**Lifetime history of needle-sharing ^b^**	
**Education**		Yes	88 (53.01%)
Illiterate	263 (10.62%)	No	78 (56.99%)
Primary and secondary education	1994 (80.56%)	**History of opioid-agonist treatments ^b^**	
University education	216 (8.82%)	Yes	125 (75.30%)
**Marital status**		No	41 (24.70%)
Married	1320 (53.33%)	**History of tattooing**	
Single	1155 (46.66%)	Yes	870 (35.15%)
**Duration of current imprisonment**		No	1605 (64.85%)
<1 year	582 (38.50%)	**Lifetime history of unprotected sex ^c^**	
≥1 year	930 (61.50%)	Yes	568 (52.40%)
**History of previous imprisonment**		No	516 (47.60%)
Yes	1512 (61.09%)		
No	963 (38.90%)		

^a^ The question was asked only from those who responded positive to “Lifetime history of drug use”. ^b^ The question was asked only from those who responded positive to “Lifetime history of drug injection”. ^c^ The question was asked only from those who had a lifetime experience of sexual contact.

**Table 2 pathogens-10-01522-t002:** Prevalence of HCV-Ab and HBsAg among Iranian incarcerated participants.

Variable	HCV-Ab				HBsAg			
	Sample Size, N	Reactive, N	Prevalence, % (95% CI)	*p*-Value ^a^	Sample Size, N	Reactive, N	Prevalence, % (95% CI)	*p*-Value ^a^
**Country region**								
North	1341	49	3.65 (2.77–4.80)	**<0.01**	1341	31	2.31 (1.63–3.26)	0.69
Centre	1134	91	8.02 (6.58–9.75)		1134	29	2.56 (1.79–3.65)	
Total	2475	140	5.66 (4.81–6.64)		2475	60	2.42 (1.89–3.11)	
**Sex**								
Male	2418	139	5.75 (4.89–6.75)	0.19	2418	59	2.44 (1.90–3.13)	0.74
Female	57	1	1.75 (0.31–9.29)		57	1	1.75 (0.31–9.29)	
**Age group**								
<30	491	8	1.63 (0.83–3.18)	**<0.01**	491	8	1.63 (0.83–3.18)	0.20
≥30	1984	132	6.65 (5.64–7.84)		1984	52	2.62 (2.00–3.42)	
**Educational level**								
Illiterate	263	18	6.84 (4.37–10.56)	**<0.01**	263	12	4.56 (2.63–7.80)	0.04
Primary and secondary education	1994	120	6.01 (5.06–7.15)		1994	45	2.26 (1.70–3.01)	
University education	216	2	0.93 (0.25–3.31)		216	3	1.39 (0.47–4.00)	
**Marital status**								
Married	1320	45	3.41 (2.56–4.53)	**<0.01**	1320	35	2.65 (1.91–3.67)	0.43
Single	1155	95	8.23 (6.78–9.95)		1155	25	2.16 (1.47–3.18)	
**Duration of current imprisonment**								
<1 year	582	18	3.09 (1.97–4.84)	**<0.01**	582	10	1.72 (0.94–3.13)	0.40
≥1 year	930	107	11.51 (9.61–14.37)		930	22	2.37 (1.57–3.56)	
**History of previous imprisonment**								
Yes	1512	125	8.27 (6.98–9.76)	**<0.01**	1512	32	2.12 (1.50–2.97)	0.21
No	963	15	1.56 (0.95–2.55)		963	28	2.91 (2.02–4.17)	
**Lifetime history of drug use**								
Yes	1429	128	8.96 (7.58–10.55)	**<0.01**	1429	36	2.52 (1.83–3.47)	0.44
No	1046	12	1.15 (0.66–1.99)		1046	24	2.29 (1.55–3.39)	
**Age of the drug use onset** ** ^b^ **								
<20	649	84	12.94 (10.58–15.75)	**<0.01**	649	14	2.16 (1.29–3.59)	0.42
≥20	780	44	5.64 (4.23–7.49)		780	22	2.82 (1.87–4.23)	
**Lifetime history of drug injection ^b^**								
Yes	166	83	50 (42.48–57.52)	**<0.01**	166	5	3.01 (1.29–6.86)	0.43
No	1258	45	3.57 (2.72–4.83)		1258	31	2.46 (1.74–3.48)	
**Lifetime history of needle-sharing ^c^**								
Yes	88	49	55.69 (42.19–63.71)	0.27	88	3	3.75 (1.03–6.02)	0.81
No	78	34	43.59 (33.14–54.64)		78	2	2.56 (0.71–8.88)	
**History of opioid-agonist treatments ^c^**								
Yes	125	69	55.20 (44.80–62.91)	**0.04**	125	2	1.60 (0.45–5.01)	0.37
No	41	14	34.15 (21.56–49.45)		41	3	7.32 (2.52–19.43)	
**History of tattooing**								
Yes	870	92	10.57 (8.70–12.79)	**<0.01**	870	20	2.30 (1.49–3.52)	0.77
No	1605	48	2.99 (2.26–3.94)		1605	40	2.49 (1.84–3.38)	
**Lifetime history of unprotected sex ^d^**								
Yes	568	49	8.63 (6.59–11.22)	0.17	568	12	2.11 (1.21–3.66)	0.55
No	516	33	6.40 (4.59–8.85)		516	13	1.83 (0.82–4.70)	

^a^ Result from chi-square test. ^b^ The question was asked only from those who responded positive to “Lifetime history of drug use”. ^c^ The question was asked only from those who responded positive to “Lifetime history of drug injection”. ^d^ The question was asked only from those who had a lifetime experience of sexual contact.

**Table 3 pathogens-10-01522-t003:** Logistic regression models to identify potential variables of HCV-Ab and HBsAg prevalence among Iranian incarcerated individuals.

Variables	HCV-Ab				HBsAg			
	Unadjusted OR (95% CI)	*p*-Value	Adjusted OR (95% CI)	*p*-Value	Unadjusted OR (95% CI)	*p*-Value	Adjusted OR (95% CI)	*p*–Value
**Sex**								
Female	1		1		1		1	
Male	3.41 (0.46–24.85)	0.22	6.44 (0.80–11.17) ^a^	0.16	1.40 (0.20–10.28)	0.74	1.13 (0.19–10.67) ^a^	0.19
**Age groups (years)**								
<30	1		1		1		1	
≥30	**4.30 (2.09–8.84)**	**<0.01**	**6.31 (3.03–13.14) ^b^**	**<0.01**	1.62 (0.76–3.44)	0.20	1.51 (0.70–3.28) ^b^	0.29
**Educational level**								
Illiterate	1		1		1		1	
Primary and secondary education	1.04 (0.55–1.97)	0.89	0.93 (0.49–1.79) ^c^	0.84	**0.36 (0.18–0.72)**	**< 0.01**	**0.35 (0.18–0.70) ^c^**	**< 0.01**
University education	**0.15 (0.03–0.68)**	**0.01**	**0.14 (0.03–0.65) ^c^**	**0.01**	**0.22 (0.06–0.82)**	**0.02**	**0.22 (0.05–0.78) ^c^**	**0.02**
**Marital status**								
Single	1		1		1		1	
Married	**0.39 (0.27–0.56)**	**<0.01**	**0.30 (0.21–0.44) ^d^**	**<0.01**	1.23 (0.73–2.06)	0.43	1.10 (0.64–1.88) ^d^	0.72
**Duration of current imprisonment**								
<1 year	1		1		1		1	
≥1 year	**4.24 (2.55–7.06)**	**<0.01**	**3.87 (2.31–6.45) ^e^**	**<0.01**	1.38 (0.65–2.94)	0.39	1.27 (0.59–2.73) ^e^	0.25
**History of previous imprisonment**								
No	1		1		1		1	
Yes	**5.69 (3.31–9.80)**	**< 0.01**	**4.63 (2.67–8.02) ^e^**	**< 0.01**	0.72 (0.43–1.20)	0.21	0.68 (0.40–1.15) ^e^	0.15
**Lifetime history of drug use**								
No	1		1		1		1	
Yes	**8.47 (4.66–15.40)**	**< 0.01**	**6.45 (3.52–11.82) ^e^**	**< 0.01**	1.10 (0.65–1.90)	0.72	1.06 (0.62–1.82) ^e^	0.81
**Age of the drug use onset**								
<20	1		1		1		1	
≥20	**0.40 (0.27–0.58)**	**<0.01**	**0.35 (0.24–0.53) ^e^**	**<0.01**	1.31 (0.66–2.60)	0.42	1.07 (0.52–2.18) ^e^	0.84
**Lifetime history of drug injection**								
No	1		1		1		1	
Yes	**24.57 (16.14–37.38)**	**<0.01**	**22.91 (14.92–35.18) ^e^**	**<0.01**	1.18 (0.45–3.08)	0.72	1.15 (0.43–3.06) ^e^	0.77
**Lifetime history of needle-sharing**								
No	1		1		1		1	
Yes	1.48 (0.41–5.28)	0.53	1.46 (0.40–5.26) ^e^	0.50	1.68 (0.30–9.47)	0.55	1.62 (0.29–9.17) ^e^	0.49
**History of opioid-agonist treatments**								
No	1		1		1		1	
Yes	**2.15 (1.03–4.45)**	**0.04**	**2.40 (1.12–5.15) ^e^**	**0.02**	0.19 (0.03–1.21)	0.08	0.16 (0.02–1.09) ^e^	0.07
**History of tattooing**								
No	1		1		1		1	
Yes	**3.83 (2.67–5.49)**	**<0.01**	**3.80 (2.62–5.51) ^e^**	**<0.01**	0.92 (0.53–1.58)	0.65	0.96 (0.55–1.68) ^e^	0.90
**Lifetime history of unprotected sex**								
No	1		1		1		1	
Yes	**1.60 (1.00–2.52)**	**0.04**	**1.72 (1.14–2.61) ^e^**	**0.01**	1.23 (0.75–2.60)	0.55	1.37 (0.61–3.11) ^e^	0.43

OR: odds ratio, CI: confidence interval, ^a^ Adjusted variable: age, educational level, marital status. ^b^ Adjusted variables: gender, educational level, marital status. ^c^ Adjusted variables: age, gender, marital status. ^d^ Adjusted variables: age, gender, educational level. ^e^ Adjusted variables: age, gender, educational level, and marital status.

**Table 4 pathogens-10-01522-t004:** Potential factors on the positivity of HCV-RNA in Iranian incarcerated participants.

Variables	HCV-RNA Positive and HCV-Ab Positive (n = 84)	HCV-RNA Negative and HCV-Ab Negative (n = 2331)	Crude Odds Ratio (% 95 CI)	*p*-Value	Adjusted Odds Ratio (% 95 CI)	*p*-Value
**Sex**						
Female	1 (1.19%)	56 (2.40%)	1		1	
Male	83 (98.81%)	2275 (97.60%)	2.04 (0.27–14.93)	0.48	2.00 (0.15–12.88) ^a^	0.46
**Age groups (years)**						
<30	3 (3.57%)	482 (20.68%)	1		1	
≥30	81 (96.43%)	1849 (79.32%)	**7.03 (2.21–22.37)**	**<0.01**	**6.90 (2.16–21.99) ^b^**	**<0.01**
**Educational level**						
Illiterate	7 (8.33%)	178 (7.63%)	1		1	
Primary and secondary education	75 (89.29%)	1939 (83.18%)	1.00 (0.44–2.14)	0.50	0.90 (0.40–2.02) ^c^	0.31
University education	2 (2.38%)	214 (9.19%)	0.23 (0.04–1.14)	0.58	0.21 (0.04–1.07) ^c^	0.06
**Marital status**						
Single	53 (63.10%)	1059 (45.43%)	1		1	
Married	31 (36.90%)	1272 (54.57%)	**0.48 (0.31–0.76)**	**<0.01**	**0.20 (0.24–0.60) ^d^**	**<0.01**
**Duration of current imprisonment**						
<1 year	10 (1.74%)	564 (40.78%)	1		1	
≥1 year	62 (86.11%)	819 (59.22%)	**4.26 (2.17–8.39)**	**<0.01**	**3.50 (1.76–6.95) ^d^**	**<0.01**
**History of previous imprisonment**						
No	14 (20.0%)	924 (39.41%)	1		1	
Yes	56 (80.0%)	1407 (60.59%)	1.28 (0.80–2.03)	0.35	1.24 (0.78–1.98) ^e^	0.28
**Lifetime history of drug use**						
No	12 (21.05%)	1004 (43.07%)	1		1	
Yes	45 (78.95%)	1327 (56.93%)	**2.80 (1.47–5.32)**	**<0.01**	**2.81 (1.47–5.26) ^e^**	**<0.01**
**Age of the drug use onset**						
<20	48 (63.16%)	564 (43.45%)	1		1	
≥20	28 (36.84%)	734 (56.55%)	**0.44 (0.27–0.72)**	**<0.01**	**0.37 (0.23–0.61) ^e^**	**<0.01**
**Lifetime history of drug injection**						
No	27 (35.53%)	1209 (93.14%)	1		1	
Yes	49 (64.47%)	89 (6.86%)	**24.65 (14.70–41.33)**	**<0.01**	**23.51 (13.86–39.87) ^e^**	**<0.01**
**Lifetime history of needle-sharing**						
No	15 (31.25%)	63 (53.38%)	1		1	
Yes	33 (68.75%)	55 (46.62%)	1.48 (0.34–6.43)	0.78	1.22 (0.27–5.48) ^e^	0.59
**History of opioid-agonist treatments**						
No	10 (20.41%)	31 (23.57%)	1		1	
Yes	39 (79.59%)	86 (76.43%)	0.30 (0.07–1.29)	0.10	0.30 (0.06–1.30) ^e^	0.10
**History of tattooing**						
No	36 (42.86%)	1553 (66.62%)	1		1	
Yes	48 (57.14%)	778 (33.38%)	**2.66 (1.70–4.13)**	**<0.01**	**2.70 (1.72–4.24) ^e^**	**<0.01**
**Lifetime history of unprotected sex**						
No	10 (48.72%)	490 (52.54%)	1		1	
Yes	17 (51.28%)	541 (47.46%)	1.36 (0.71–1.57)	0.28	1.20 (0.73–1.55) ^e^	0.22

OR: odds ratio, CI: confidence interval, ^a^ Adjusted variable: age, educational level, marital status. ^b^ Adjusted variables: gender, educational level, marital status. ^c^ Adjusted variables: age, gender, marital status. ^d^ Adjusted variables: age, gender, educational level. ^e^ Adjusted variables: age, gender, educational level, and marital status.

**Table 5 pathogens-10-01522-t005:** Factors modifying chronicity of HCV infection among Iranian incarcerated individuals.

Variables	HCV-RNA Positive and HCV-Ab Positive (n = 84)	HCV-RNA Negative and HCV-Ab Positive (n = 30)	Crude Odds Ratio (% 95 CI)	*p*-Value	Adjusted Odds Ratio (% 95 CI)	*p*-Value
**Sex**						
Female	1 (1.19%)	0 (0%)	1		1	
Male	83 (98.81%)	30 (100%)	-	-	-	-
**Age groups (years)**						
<30	3 (3.57%)	5 (16.67%)	1		1	
≥30	81 (96.43%)	25 (83.33%)	**5.40 (1.20–24.20)**	**0.04**	**5.24 (1.00–27.30) ^b^**	**0.02**
**Educational level**						
Illiterate	7 (8.33%)	3 (10%)	1		1	
Primary and secondary education	75 (89.29%)	27 (90%)	1.19 (0.28–4.93)	0.81	1.26 (0.28–5.63) ^c^	0.31
University education	2 (2.38%)	0 (0%)	-	-	-	-
**Marital status**						
Single	53 (63.10%)	22 (73.33%)	1		1	
Married	31 (36.90%)	8 (26.67%)	1.60 (0.63–4.04)	0.31	1.53 (0.59–3.97) ^d^	0.37
**Duration of current imprisonment**						
<1 year	10 (1.74%)	3 (23.08%)	1		1	
≥1 year	62 (86.11%)	26 (76.92%)	**1.39 (1.17–10.39)**	**0.05**	**1.43 (1.04–8.44) ^d^**	**0.04**
**History of previous imprisonment**						
No	28 (33.33%)	10 (33.33%)	1		1	
Yes	56 (66.67%)	20 (66.67%)	1.00 (0.41–2.42)	0.89	1.12 (0.44–2.81) ^e^	0.75
**Lifetime history of drug use**						
No	12 (21.05%)	8 (42.12%)	1			
Yes	45 (78.95%)	22 (57.88%)	1.36 (0.48–3.81)	0.55	1.62 (0.55–3.76) ^e^	0.38
**Age of the drug use onset**						
<20	48 (63.16%)	20 (71.43%)	1		1	
≥20	28 (36.84%)	8 (28.57%)	0.69 (0.27–1.77)	0.54	0.66 (0.24–1.80) ^e^	0.43
**Lifetime history of drug injection**						
No	27 (35.53%)	8 (27.59%)	1		1	
Yes	49 (64.47%)	21 (72.41%)	0.69 (0.27–1.77)	0.44	0.66 (0.24–1.80) ^e^	0.43
**Lifetime history of needle-sharing**						
No	15 (31.25%)	8 (34.78%)	1		1	
Yes	33 (68.75%)	15 (65.22%)	1.17 (0.40–3.36)	0.76	1.23 (0.41–2.98) ^e^	0.70
**History of opioid-agonist treatments**						
No	10 (20.41%)	3 (14.29%)	1		1	
Yes	39 (79.59%)	18 (85.71%)	0.75 (0.15–2.65)	0.82	0.70 (0.16–2.91) ^e^	0.62
**History of tattooing**						
No	36 (42.86%)	7 (23.33%)	1		1	
Yes	48 (57.14%)	23 (76.67%)	1.14 (0.39–3.33)	0.80	1.27 (0.41–3.87) ^e^	0.67
**Lifetime history of unprotected sex**						
No	10 (37.04%)	9 (60%)	1		1	
Yes	17 (62.96%)	6 (40%)	1.11 (0.45–2.73)	0.85	1.20 (0.69–1.98) ^e^	0.43

OR: odds ratio, CI: confidence interval, ^b^ Adjusted variables: gender, educational level, marital status. ^c^ Adjusted variables: age, gender, marital status. ^d^ Adjusted variables: age, gender, educational level. ^e^ Adjusted variables: age, gender, educational level, and marital status.

## Data Availability

The data that support the findings of this study are available from the corresponding authors upon reasonable request.

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
