# Peer review of "Prevalence of Hepatitis B and Hepatitis C Infections among Incarcerated Individuals in Iran: A Cross-Sectional National Bio-behavioral Study in 2019"

_pathogens, 2021, doi:10.3390/pathogens10111522_

Round 1

Reviewer 1 Report

In the introduction, the authors need to briefly introduce the major conclusions from the first and second national surveys performed in 2015 and 2016 and provide a more reasonable rationale for why the first survey was performed.

In line 153, the prevalence of HCV-Ab was higher in men than in woman group, but the difference was not statistically significant according to the table. 2. Thus, it seems not appropriate to state the conclusion if not significant, or the authors need to rephrase the sentence to emphasize or discuss why the difference was not significant. The same issue in lines 173-179 for HBV prevalence with different variables, the discussed geography, and drug use were not significant variables, and educational disparities that were significantly correlated with the prevalence were not discussed here.

In the current version, there was no further evidence that the 84 prisoners with positive HCV RNA were chronically infected. Please provide more details.

Line 166, typo error “participant..s.”, line 194 “age>20” should be “age ≥20”. “OR” full name should be detailed in the main text or table legends.

Author Response

  • In the introduction, the authors need to briefly introduce the major conclusions from the first and second national surveys performed in 2015 and 2016 and provide a more reasonable rationale for why the first survey was performed.

REPLY> Thanks for your comment, we added the brief findings of these studies in the introduction section and highlighted them in the revised manuscript.

  • In line 153, the prevalence of HCV-Ab was higher in men than in woman group, but the difference was not statistically significant according to the table. 2. Thus, it seems not appropriate to state the conclusion if not significant, or the authors need to rephrase the sentence to emphasize or discuss why the difference was not significant. The same issue in lines 173-179 for HBV prevalence with different variables, the discussed geography, and drug use were not significant variables, and educational disparities that were significantly correlated with the prevalence were not discussed here.

REPLY> Thanks for your comment, we rephrased the sentences to present that these differences were not statistically significant and highlighted them in the revised manuscript.

  • In the current version, there was no further evidence that the 84 prisoners with positive HCV RNA were chronically infected. Please provide more details.

REPLY> Chronic HCV infection has a definition of being positive for HCV infection for more than 6 months (positive for HCV-Ab and HCV-RNA). This study could not identify the chronic status of HCV infection in a cross-sectional setting and we preferred to refer to it as “current HCV infection”.

  • Line 166, typo error “participants.”, line 194 “age>20” should be “age ≥20”. “OR” full name should be detailed in the main text or table legends.

REPLY> Thanks for your comment, we edited and highlighted all typos and added abbreviations.

Reviewer 2 Report

AUTHORS

Title: Prevalence of hepatitis B and hepatitis C infections among incarcerated individuals in Iran: A cross-sectional national bio-behavioral study in 2019

Authors aimed at determining the prevalence of hepatitis B and C infections among incarcerated individuals in five provinces of Iran, 2019. The manuscript is interesting and discloses novel data, using a large dataset (2500 individuals), which is the truly novel aspect. The potential impact on infectious diseases control and monitoring is high and the manuscript sees to warrant sufficient robustness. I do have a few minor questions/comments that could help improve it.

Why did authors HBsAg and HCV-Ab and then checked for HCV RNA?

“using AmpliSens® HCV-FRT PCR kit” – place manufacturer city/country

Kolmogorov Smirnov test is known for not being the best approach for testing normality (see RB D'Agostino, "Tests for Normal Distribution" in Goodness-Of-Fit Techniques edited by RB D'Agostino and MA Stephens, Macel Dekker, 1986) and other approaches such as D'Agostino-Pearson are likely to be more robust. Please confirm your initial assessment with another (more robust) test.

For your logistic regression, why dichotomize in below and above 30 years old?

The same for Age of the drug use onset (20 years) and duration of imprisonment (1 year)

Author Response

  • Why did authors HBsAg and HCV-Ab and then checked for HCV RNA?

REPLY> While HCV-Ab testing is a serological marker of HCV exposure, used in epidemiological studies, it is a screening biomarker used for the diagnosis of HCV infection. HCV-Ab is an inexpensive and reliable biomarker for screening of HCV infection followed by evaluation of HCV-RNA in HCV-Ab reactive cases.

  • “Using AmpliSens® HCV-FRT PCR kit” – place manufacturer city/country

REPLY> The manufacturer of this kit is Russian. We added and highlighted it to the manuscript.

  • Kolmogorov Smirnov test is known for not being the best approach for testing normality (see RB D'Agostino, "Tests for Normal Distribution" in Goodness-Of-Fit Techniques edited by RB D'Agostino and MA Stephens, Macel Dekker, 1986) and other approaches such as D'Agostino-Pearson are likely to be more robust. Please confirm your initial assessment with another (more robust) test.

REPLY> Thanks for your comment. We used also the Shapiro Wilk test to initial data assessment. This test is added and highlighted in the statistical analysis section in the revised manuscript.

  • For your logistic regression, why dichotomize in below and above 30 years old?

REPLY> This type of classification has been done due to the time of hepatitis B vaccination in Iran. With this category, people over 30 have been vaccinated for hepatitis B and people under 30 haven’t been vaccinated.

  • The same for Age of the drug use onset (20 years) and duration of imprisonment (1 year).

REPLY> Thanks, this classification has been done based on the literature review and previous study results. Studies have shown that, for many adolescents in Iran, the age of drug use onset is under age 18-20, and many of them used drugs for the first time in schools or friends’ homes. Nearly half of drug-using university students in one study had been familiar with drugs since their adolescence.

Reference: Students' attitudes and practices towards drug and alcohol use at Tabriz University of Medical Sciences. (Jodati AR, Shakurie SK, Nazari M, Raufie MB; East Mediterr Health J. 2007 Jul-Aug; 13(4):967-71)

Reference: Substance abuse among Iranian high school students. (Momtazi S, Rawson RA. Current opinion in psychiatry. 2010 May;23(3):221.)

In Iran, most prisoners do not have serious crimes and the length of their imprisonment is often short, so the category duration of imprisonment was selected based on below and above one year.

Round 2

Reviewer 1 Report

No more concerns.

Reviewer 2 Report

Authors have adequately addressed this reviewers concerns and the manuscript is now fit for publication.